# Determinants of individuals' belief in fake news: A scoping review determinants of belief in fake news

**Kirill Bryanov**[ID]*, **Victoria Vziatysheva**[ID]

Laboratory for Social and Cognitive Informatics, National Research University Higher School of Economics, St. Petersburg, Russia

* kbryanov@hse.ru

**Data Availability Statement:** All relevant data are available within the paper. Search protocol is described in the text, and Table 3 contains information about all studies included in the review.

**Funding:** The research was supported by the Russian Scientific Fund Grant № 19-18-00206

## Abstract

### Background

Proliferation of misinformation in digital news environments can harm society in a number of ways, but its dangers are most acute when citizens believe that false news is factually accurate. A recent wave of empirical research focuses on factors that explain why people fall for the so-called fake news. In this scoping review, we summarize the results of experimental studies that test different predictors of individuals' belief in misinformation.

### Methods

The review is based on a synthetic analysis of 26 scholarly articles. The authors developed and applied a search protocol to two academic databases, Scopus and Web of Science. The sample included experimental studies that test factors influencing users' ability to recognize fake news, their likelihood to trust it or intention to engage with such content. Relying on scoping review methodology, the authors then collated and summarized the available evidence.

### Results

The study identifies three broad groups of factors contributing to individuals' belief in fake news. Firstly, message characteristics—such as belief consistency and presentation cues—can drive people's belief in misinformation. Secondly, susceptibility to fake news can be determined by individual factors including people's cognitive styles, predispositions, and differences in news and information literacy. Finally, accuracy-promoting interventions such as warnings or nudges priming individuals to think about information veracity can impact judgements about fake news credibility. Evidence suggests that inoculation-type interventions can be both scalable and effective. We note that study results could be partly driven by design choices such as selection of stimuli and outcome measurement.

### Conclusions

We call for expanding the scope and diversifying designs of empirical investigations of people's susceptibility to false information online. We recommend examining digital platforms

(2019–2021) at the National Research University Higher School of Economics. Funder website: https://grant.rscf.ru/enexp/ The funders had no role in study design, data collection and analysis, decision to publish, or preparation of the manuscript.

**Competing interests:** The authors have declared that no competing interests exist.

beyond Facebook, using more diverse formats of stimulus material and adding a comparative angle to fake news research.

## Introduction

Deception is not a new phenomenon in mass communication: people had been exposed to political propaganda, strategic misinformation, and rumors long before much of public communication migrated to digital spaces [1]. In the information ecosystem centered around social media, however, digital deception took on renewed urgency, with the 2016 U.S. presidential election marking the tipping point where the gravity of the issue became a widespread concern [2, 3]. A growing body of work documents the detrimental effects of online misinformation on political discourse and people's societally significant attitudes and beliefs. Exposure to false information has been linked to outcomes such as diminished trust in mainstream media [4], fostering the feelings of inefficacy, alienation, and cynicism toward political candidates [5], as well as creating false memories of fabricated policy-relevant events [6] and anchoring individuals' perceptions of unfamiliar topics [7].

According to some estimates, the spread of politically charged digital deception in the buildup to and following the 2016 election became a mass phenomenon: for example, Allcott and Gentzkow [1] estimated that the average US adult could have read and remembered at least one fake news article in the months around the election (but see Allen et al. [8] for an opposing claim regarding the scale of the fake news issue). Scholarly reflections upon this new reality sparked a wave of research concerned with a specific brand of false information, labelled *fake news* and most commonly conceptualized as non-factual messages resembling legitimate news content and created with an intention to deceive [3, 9]. One research avenue that has seen a major uptick in the volume of published work is concerned with uncovering the factors driving people's ability to discern fake from legitimate news. Indeed, in order for deceitful messages to exert the hypothesized societal effects—such as catalyzing political polarization [10], distorting public opinion [11], and promoting inaccurate beliefs [12]—the recipients have to believe that the claims these messages present are true [13]. Furthermore, research shows that the more people find false information encountered on social media credible, the more likely they are to amplify it by sharing [14]. The factors and mechanisms underlying individuals' judgements of fake news' accuracy and credibility thus become a central concern for both theory and practice.

While message credibility has been a longstanding matter of interest for scholars of communication [15], the post-2016 wave of scholarship can be viewed as distinct on account of its focus on particular news formats, contents, and mechanisms of spread that have been prevalent amid the recent fake news crisis [16]. Furthermore, unlike previous studies of message credibility, the recent work is increasingly taking a turn towards developing and testing potential solutions to the problem of digital misinformation, particularly in the form of interventions aimed at improving people's accuracy judgements.

Some scholars argue that the recent rise of fake news is a manifestation of a broader ongoing epistemological shift, where significant numbers of online information consumers move away from the standards of evidence-based reasoning and pursuit of objective truth toward "alternative facts" and partisan simplism—a malaise often labelled as the state of "post-truth" [17, 18]. Lewandowsky and colleagues identify large-scale trends such as declining social capital, rising economic inequality and political polarization, diminishing trust in science, and an increasingly fragmented media landscape as the processes underlying the shift toward the

"post-truth." In order to narrow the scope of this report, we specifically focus on the news media component of the larger "post-truth" puzzle. This leads us to consider only the studies that explore the effects of misinformation packaged in news-like formats, perforce leaving out investigations dealing with other forms of online deception–for example, messages coming from political figures and parties [19] or rumors [20].

The apparently vast amount and heterogeneity of recent empirical research output addressing the antecedents to people's belief in fake news calls for integrative work summarizing and mapping the newly generated findings. We are aware of a single review article published to date synthesizing empirical findings on the factors of individuals' susceptibility to believing fake news in political contexts, a narrative summary of a subset of relevant evidence [21]. In order to systematically survey the available literature in a way that permits both transparency and sufficient conceptual breadth, we employ a scoping review methodology, most commonly used in medical and public health research. This method prescribes specifying a research question, search strategy, and criteria for inclusion and exclusion, along with the general logic of charting and arranging the data, thus allowing for a transparent, replicable synthesis [22]. Because it is well-suited for identifying diverse subsets of evidence pertaining to a broad research question [23], scoping review methodology is particularly relevant to our study's objectives. We begin our investigation with articulating the following research questions:

RQ1: What factors have been found to predict individuals' belief in fake news and their capacity to discern between false and real news?

RQ2: What interventions have been found to reduce individuals' belief in fake news and boost their capacity to discern between false and real news?

In the following sections, we specify our methodology and describe the findings using an inductively developed framework organized around groups of factors and dependent variables extracted from the data. Specifically, we approached the analysis without a preconceived categorization of the factors in mind. Following our assessment of the studies included in the sample, we divided them into three groups based on whether the antecedents of belief in fake news that they focus on 1) reside within the individual or 2) are related to the features of the message, source, or information environment or 3) represent interventions specifically designed to tackle the problem of online misinformation. We conclude with a discussion of the state of play in the research area under review, identify strengths and gaps in existing scholarship, and offer potential avenues for further advancing this body of knowledge.

## Materials and methods

Our research pipeline has been developed in accordance with PRISMA guidelines for systematic scoping reviews [24] and contains the following steps: a) development of a review protocol; b) identification of the relevant studies; c) extraction and charting of the data from selected studies, elaboration of the emerging themes; d) collation and summarization of the results; e) assessment of the strengths and limitations of the body of literature, identification of potential paths for addressing the existing gaps and theory advancement.

### Search strategy and protocol development

At the outset, we defined the target population of texts as English-language scholarly articles published in peer-reviewed journals between January 1, 2016 and November 1, 2020 and using experimental methodology to investigate the factors underlying individuals' belief in false news. We selected this time frame with the intention to specifically capture the research output that emerged in response to the "post-truth" turn in the public and scholarly discourse that

many observers link to the political events of 2016, most notably Donald Trump's ascent to U. S. presidency [17]. Because we were primarily interested in causal evidence for the role of various antecedents to fake news credibility perceptions, we decided to focus on experimental studies. Our definition of experiment has been purposefully lax, since we acknowledged the possibility that not all relevant studies could employ rigorous experimental design with random assignment and a control group. For example, this would likely be the case for studies testing factors that are more easily measured than manipulated, such as individual psychological predispositions, as predictors of fake news susceptibility. We therefore included investigations where researchers varied at least one of the elements of news exposure: Either a hypothesized factor driving belief in fake news (both between or within subjects), or veracity of news used as a stimulus (within-subjects). Consequently, the studies included in our review presented both causal and correlational evidence.

Upon the initial screening of relevant texts already known to the authors or discovered through cross-referencing, it became apparent that proposed remedies and interventions enhancing news accuracy judgements should also be included into the scope of the review. In many cases practical solutions are presented alongside fake news believability factors, while in several instances testing such interventions is the reports' primary concern. We began with developing the string of search terms informed by the language found in the titles of the already known relevant studies [14, 25–27], then enhanced it with plausible synonymous terms drawn from the online service *Thesaurus.com*. As the initial version of this report went into peer review, we received reviewer feedback suggesting that some of the relevant studies, particularly on the topic of inoculation-based interventions, were left out. We modified our search query accordingly, adding further three inoculation-related terms. The ultimate query looked as follows:

(belie* OR discern* OR identif* OR credib* OR evaluat* OR assess* OR rating OR

rate OR suspic* OR "thinking" OR accura* OR recogn* OR susceptib* OR malleab* OR trust* OR resist* OR immun* or innocul*) AND (false* OR fake OR disinform* OR misinform*).

Based on our understanding that the relevant studies should fall within the scope of such disciplines as media and communication studies, political science, psychology, cognitive science, and information sciences, we identified two citation databases, Scopus and Web of Science, as the target corpora of scholarly texts. Web of Science and Scopus are consistently ranked among leading academic databases providing citation indexing [28, 29]. Norris and Oppenheim [30] argue that in terms of record processing quality and depth of coverage these databases provide valid instruments for evaluating scholarly contributions in social sciences. Another possible alternative is Google Scholar, which also provides citation indexing and is often considered the largest academic database [31]. Yet, according to some appraisals, this database lacks quality control [32], transparency, and can contribute to parts of relevant evidence being overlooked when used in systematic reviews [33]. Thus, for the purposes of this paper, we chose WoS and Scopus as sources of data.

## Relevance screening and inclusion/exclusion criteria

Using title search, our queries resulted in 1622 and 1074 publications in Scopus and Web of Science, respectively. The study selection process is demonstrated in Fig 1.

We began the search with crude title screening performed by the authors (KB and VV) on each database independently. On this stage, we mainly excluded obviously irrelevant articles (e.g. research reports mentioning false-positive biochemical tests results) and those whose titles unambiguously indicated that the item was outside of our original scope, such as work in

the field of machine learning on automated fake news detection. Both authors' results were then cross-checked, and disagreements resolved. This stage narrowed our selection down to 109 potentially relevant Scopus articles and 76 WoS articles. Having removed duplicate items present in both databases, we arrived at the list of 117 unique articles retained for abstract review.

On the abstract screening stage, we excluded items that could be identified as utilizing non-experimental research designs. Furthermore, on this stage we determined that all articles that fit our intended scope include at least one of the following outcome variables: 1) perceived credibility, believability, or accuracy of false news messages and 2) a measure of the capacity to discern false from authentic news. Screening potentially eligible abstracts suggested that studies not addressing one of these two outcomes do not answer the research questions at the center of our study. Seventy articles were thus removed, leaving us with 45 articles for full-text review.

The remaining articles were read in full by the authors independently, disagreements on whether specific items fit the inclusion criteria resolved, resulting in the final sample of 26 articles (see Table 1 for the full list of included studies). Since our primary focus is on perceptions of false media content and corresponding interventions designed to improve news delivery and consumption practices, we only included the experiments that utilized a news-like format of the stimulus material. As a result, we forwent investigations focusing on online rumors, individual politicians' social media posts, and other stimuli that were not meant to represent content produced by a news organization. We did not limit the range of platforms where the news articles were presented to participants, since many studies simulated the processes of news selection and consumption in high-choice environments such as social media feeds. We then charted the evidence contained therein according to a categorization based on the outcome and independent variables that the included studies investigate.

## Results

### Outcome variables

Having arranged the available evidence along a number of ad-hoc dimensions, including the primary independent variables/correlates and focal outcome variables, we opted for a presentation strategy that opens with a classification of study dependent variables. Our analysis revealed that the body of scholarly literature under review is characterized by a significant heterogeneity of outcome variables. The concepts central to our synthesis are operationalized and measured in a variety of ways across studies, which presents a major hindrance to comparability of their results. In addition, in the absence of established terminology these variables are often labelled differently even when they represent similar constructs.

In addition to several variations of the dependent variables that we used as one of the inclusion criteria, we discovered a range of additional DVs relevant to the issue of online misinformation that the studies under review explored. The resulting classification is presented in Table 2 below.

As visible from Table 2, the majority of studies in our sample measured the degree to which participants identified news messages or headlines as credible, believable or accurate. This strategy was utilized in experiments that both exposed individuals to made-up messages only, and those where stimulus material included a combination of real and fake items. Studies of the former type examined the effects of message characteristics or presentation cues on perceived credibility of misinformation, while the latter stimulus format also enabled scholars to examine the factors driving the accuracy of people's identification of news as real or fake. In most instances, these synthetic "media truth discernment" scores were constructed post-hoc

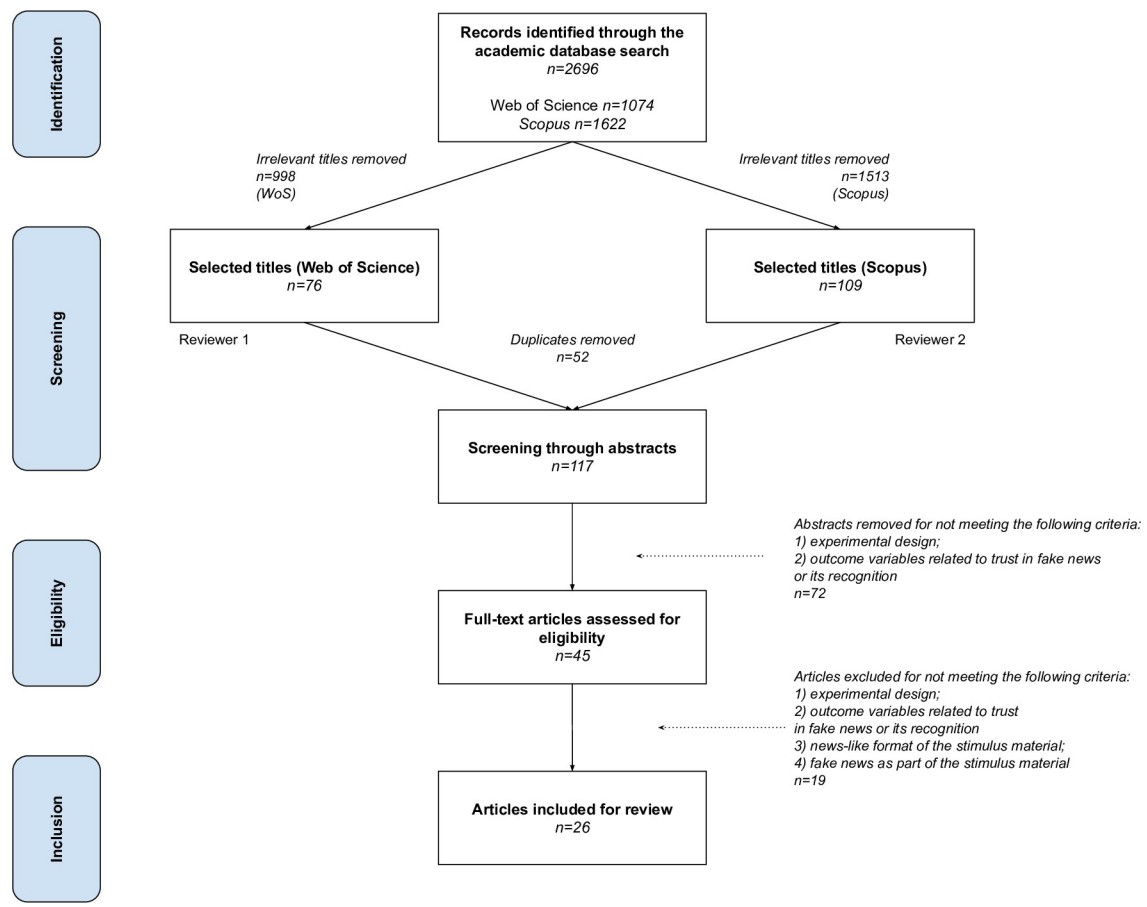

**Fig 1. Flow diagram of the study selection process.**

by matching participants' credibility responses to the known "ground truth" of messages that they were asked to assess. These individual discernment scores could then be matched with the respondent's or message's features to infer the sources of systematic variation in the aggregate judgement accuracy.

Looking at credibility perceptions of real and false news separately also enabled scholars to determine whether the effects of factors or interventions were symmetric for both message types. In a media environment where the overwhelming majority of news is real after all [27], it is essential to ensure both that fake news is dismissed, and high-quality content is trusted.

Another outcome that several studies in our sample investigated is the self-reported likelihood to share the message on social media. Given that social platforms like Facebook are widely believed to be responsible for the rapid spread of deceitful political content in recent years [2], the determinants of sharing behavior are central to developing effective measures for limiting the reach of fake news. Moreover, in at least one study [34] researchers explicitly used sharing intent as a proxy for a news accuracy judgement in order to estimate perceived accuracy without priming participants' thinking about veracity of information. This approach appears promising given that this as well as other studies reported sizable correlations between perceived accuracy and sharing intent [35–37], yet it is obviously limited as a host of considerations beyond credibility can inform the decision to share a news item on social media.

 

**Table 1. Key characteristics of included articles.**

| Author(s), year | Factors | Sample | Procedures and stimuli* | Outcome variables & measurement | Results |
|---|---|---|---|---|---|
| Clayton et al. 2019a [45] | Message source cue (concordant, discordant, no source) | N = 3932; MTurk, U.S. | Political preferences measures → Random assignment to a version of an article excerpt on health care reform following a 2 (true/false) x 3 (attribution to CNN/Fox/no attribution) design → Outcome measurement. | Accuracy of a *false* statement mirroring the content of the article in the false condition (1–4) | Regardless of their political preferences, individuals exposed to false information were significantly more likely to rate the false statement as accurate than individuals exposed to true information. This does not vary by partisanship or ideology of the participant or source. |
| Mena et al. 2020 [46] | 1) Trusted endorsement; 2) Post popularity. | N = 479, Mturk, U.S., active Instagram users. | Rating trustworthiness of 5 Instagram celebrities → Random assignment to one version of an Instagram post (Non-political news headline, unknown news source) following a 2 (high number of likes/no number) x 2 (endorsement by a trustworthy celebrity/no endorsement) design → Outcome measurement. | Message credibility scale: believable, accurate, and authentic (1–7) | 1) Trusted endorsement significantly increased perceived credibility of the post; 2) Bandwagon cue did not significantly increase the perceived credibility of the post. |
| Luo et al. 2020 [40] | 1) Truth-bias; 2) News topic; 3) Bandwagon cues; 4) Trusted endorsement. | N1 = 370, Mturk, U.S.; N2 = 736, Mturk, U.S. | **Study 1:** Prompt: "*All headlines have been widely circulated on Facebook and some of them involve blatant fake content*" → Random assignment to one of the conditions following a 3 (Topic: politics/ health/science) × 2 (Veracity: real/fake) design; exposure to 5 fake and 5 real news headlines, presented as Facebook posts → Outcome measurement. **Study 2:** Random assignment to one of the conditions following a 3 (Topic: politics/science/health) × 2 (Source of Likes: friend/ user) → Exposure to 8 headlines with 4 different combinations of veracity (true/fake) and number of likes (high/low) → Outcome measurement. | Study 1: 1) Message credibility scale: believable, accurate, and authentic (1–7 scale) 2) Detection accuracy: recoded from credibility responses. Study 2: 1) The extent to which an article is fake or real (1–7 scale); 2) Detection accuracy: recoded from credibility responses. | 1) Deception bias: average credibility score significantly below the midpoint of the scale; 2) Detection accuracy across two studies significantly better/not significantly worse than chance. Health news detected more accurately than science news, but less accurately than political news; no effects of veracity on detection accuracy. 3) Number of likes increased the perceived credibility of both real and fake headlines. 4) Likes by friends did not increase perceived credibility. 5) Fake (52.6%) and real news (45.8%) detection accuracies were significantly different. |
| Kluck et al. 2019 [35] | 1) Negative comments; 2) Bandwagon cues of credibility. | N = 240, recruited via Facebook, German. | Random assignment to one of the conditions following a 3 (user comments: positive/negative/no comments) x 3 (numerical credibility rating: positive/negative/no rating) design → Exposure to a made-up news story, from fictitious source, presented as a Facebook post → Outcome measurement. | 1) Credibility scale of 8 items (1–7); 2) Willingness to share— publicly and privately separately (1–7 scale) | 1) Comments with a negative (but not positive) valence negatively affected participants' perceived article credibility relative to no comments; Negative comments reduced the willingness to share post via decreased credibility; 2) Bandwagon cues had no effect on credibility or sharing intent. |
| Schaewitz et al. 2020 [13] | 1) Message cues: sensational, subjective, inconsistent, image manipulation, dubious source; 2) Receiver individual dispositions: need for cognition, faith in intuition; 3) Receiver behavior: frequency of using verification strategies online; internet search self-efficacy; 4) Support for the news topic. | N = 294, recruited via Facebook and a crowdsourcing service, German. | Random assignment to one of the conditions following a 6 (message characteristics: baseline/sensational/ subjective/ inconsistent/ image manipulation/dubious source) x 2 (news topic: crime/healthcare) design → Exposure to a fabricated message from a fictitious news source, presented as both a Facebook post and a full article → Outcome measurement & individual dispositions battery. | 1) Message credibility scale (1–7); 2) Perceived accuracy 0–100%, using slider; 3) Source credibility; 4) Likelihood of sharing. | 1) Message characteristics did not move participants' assessments of credibility or accuracy of the message, likelihood to share, or evaluations of the source; 2) Frequency of using verification strategies and Internet search self-efficacy had no effect on outcomes of interest; 3) Need for cognition was a negative predictor for credibility and accuracy of disinformation, faith in intuition was not; 4) Support for the topic increased credibility judgements, but not perceived accuracy;. 5) Knowledge and opinion on the topic predicted sharing. |
| Moravec et al. 2019 [41] | 1) Headline's belief consistency; 2) Fake news flag. | N = 83, students a business course, U.S. | Measures of political orientation & political conservatism across topics → Exposure to 50 headlines presented as Facebook posts (either verifiably true or false), 10 topics, 20 fake news flags assigned randomly → Headline credibility evaluation with simultaneous EEG measurement. | 1) Headline credibility scale; 2) Time to credibility judgement; 3) Changes in cognition using time-frequency analysis of EEG data. | 1) Fake news flag did not affect headline credibility as it was not strong enough to overcome a priori beliefs; 2) Participants were not more likely to believe true headlines; 3) Participants were more likely to trust belief-consistent headlines. 4) Cognitive attention was directed at attitude-consistent headlines; attitude-discrepant ones were ignored. |
| Kim & Dennis 2019 [14] | 1) Presentation format: highlighting source; 2) Source reliability ratings; 3) Headline's belief consistency. | N1 = 445, recruited via Facebook & Qualtrics; N2 = 501 active Facebook users, Qualtrics panel, U.S. | **Study 1:** Pre-test → Exposure to stimuli: 12 made-up headlines presented as Facebook posts, topic: abortion; 6 left-leaning and 6 right-leaning. All treatments presented to all participants in a repeated-measures design: 1) Headline-primacy format; 2) Source-primacy format; 3) Source-primacy with source ratings → Topic importance and respondent's position on it measured after rating each headline → Outcome measurement. **Study 2:** Similar to Study 1, but with between-subjects design with a random assignment to one of 4 conditions: 1) Control; 2) Headline-primacy format; 3) Source-primacy format;4) Source primacy with source ratings. | 1) Believability of articles scale; 2) Likelihood to engage with the article: Read, like, post a supporting comment,post an opposing comment, Share. | 1) Emphasizing the source makes users believe the headline less—regardless of the source; 2) Low source ratings make users believe the headline less; more than twice the effect of presentation format; 3) Users are more likely to rate as redible, read, like, post supporting comments, and share articles that they agree with. |

*(Continued)*

 

**Table 1.** (Continued)

| Author(s), year | Factors | Sample | Procedures and stimuli* | Outcome variables & measurement | Results |
|---|---|---|---|---|---|
| Pennycook, Cannon & Rand 2018 [47] | 1) Fluency via prior exposure; 2) Headline political concordance; 3) Disputed flag. | N1 = 515; N2 = 949; N3 = 940, MTurk, U.S. | **Study 1:** Stimulus: 4 known true statements, 4 extremely implausible statements, 10 true and 10 false unknown trivia facts. Procedure: Familiarization (rating interestingness of statements) → Demographics → Accuracy of all statements measured. **Study 2:** Stimulus: Headlines presented as Facebook posts (real from mainstream sources; fake from Snopes.com). Procedure: Familiarization → Distraction (Demographics and political attitudes) → Assessment of 24 headlines (12 seen and 12 new, counterbalanced). **Study 3:** Similar to Study 2, but with a follow-up session a week after the first assessment to evaluate effects' persistence. | 1) Accuracy of the claim in the headline (4-point scale); 2) Sharing intent. | 1) No effect of repetition on perceived accuracy of patently false statements—plausibility as a boundary condition; 2) Prior exposure increased accuracy ratings of headlines, regardless of their veracity, Disputed label presence, or political consistency; 3) The effect of repetition on perceived accuracy persisted after a week and increased with an additional repetition. |
| Pennycook & Rand 2019 [25] | 1) Analytic thinking (as measured by the Cognitive Reflection Test); 2) Headline political concordance. | N = 3446, MTurk, U.S. | **Study 1:** Exposure to stimulus: Headlines presented as Facebook posts (15 real from mainstream sources; 15 fake from Snopes.com, ideologically counterbalanced) → Ooutcome measurement→ Cognitive Reflection Test. **Study 2:** Replication of Study 1 with a larger set of items and in a larger sample. **Study 3:** A post hoc analysis of the data from the first two studies: correlation between CRT and out-of-sample headline plausibility ratings. | 1) Accuracy of the claim in the headline (4-point scale); 2) Sharing intent; 3) Media truth discernment (average accuracy ratings of real news minus average accuracy ratings of fake news) —post-hoc analysis. | 1) More analytic individuals rated fake news as less accurate and real news as more accurate regardless of ideological consistency; similar pattern for sharing intent; 2) Participants were better able to discern real from fake headlines that were politically concordant; 3) Trump supporters were somewhat less accurate than Clinton supporters across all headlines; 4) Headline implausibility moderates the relationship between CRT results and perceptions of item accuracy. |
| Bago et al. 2020 [43] | 1) Conditions favoring heightened deliberation (increased time & reduced cognitive load); 2) Headline political concordance. | N = 1635, MTurk, U.S. | Random assignment to either 1) One-response condition with no constraints or 2) Two-response condition with time and cognitive load constraints → Exposure to stimulus: 16 true and false headlines presented as Facebook posts (real from mainstream sources; fake from Snopes.com, ideologically counterbalanced), time limit and additional memory load for two-response group → Outcome measurement → Only for two-response group: Outcome measurement without time and cognitive constraints, allowing for deliberation. | Accuracy of the claim in the headline (4-point scale); | Deliberation increased fake news discernment accuracy by 7–8% (slightly more for concordant headlines than discordant). |
| Martel et al. 2020 [48] | 1) Self-reported use of emotion; 2) Induced reliance on emotion. | N1 = 409 MTurk, U.S.; N2 = 3884, Mturk and Lucid, U.S. | **Study 1:** Positive and Negative Affect Schedule scale (PANAS) test → Exposure to 20 true and false headlines presented as Facebook posts (real from mainstream sources; fake from Snopes.com, ideologically counterbalanced) → Outcome measurement. **Study 2:** Random assignment to 1 of the conditions: emotion induction, reason induction, or no induction → Exposure to stimulus identical to that in Study 1 → Outcome measurement. | 1) Accuracy of the claim in the headline (4-point scale); 2) Media truth discernment | 1) Momentary emotion, regardless of the specific type or valence, predicted increased belief in fake news and decreased discernment between real and fake news; 2) Inducing emotional processing increased belief in fake news and decreased truth discernment, regardless of headline concordance. |
| Bronstein et al. 2019 [42] | 1) Delusion-like ideation; 2) Actively open-minded thinking; 3) Analytic thinking; 4) Dogmatism; 5) Religious fundamentalism. | M = 948, MTurk, U.S. | **Study 1:** Individual measures: Cognitive Reflection Test, measures for delusion-like ideation, actively open-minded thinking, dogmatism, religious fundamentalism → Exposure to stimulus: Headlines presented as Facebook posts (real from mainstream sources; fake from Snopes.com, ideologically counterbalanced) → Outcome measurement. Respondents were randomly assigned to receive either individual measures or news evaluation task first. **Study 2:** CRT, news evaluation task (identical to Study 1), other individual measures from Study 1, all presented in random order. | 1) Accuracy of the claim in the headline (4-point scale); averaged across all fake and all real stories separately ("news sensitivity"/"media truth discernment") | 1) Delusion-like ideation, dogmatism, and religious fundamentalism correlated with increased belief in fake news, but did not correlate with belief in real news (delusion-like ideation and religious fundamentalism) or negatively correlated with it (dogmatism). 2) Delusion-like ideation, dogmatism, religious fundamentalism, and belief in fake news were all negatively correlated with analytic and actively open-minded thinking; 3) Analytic and actively open-minded thinking correlated with belief in real news; 4) Delusion-like ideation, dogmatism, and religious fundamentalism were negatively correlated with fake news discernment, while actively open minded and analytic thinking were positively correlated with fake news discernment. |

(*Continued*)

**Table 1.** (Continued)

| Author(s), year | Factors | Sample | Procedures and stimuli* | Outcome variables & measurement | Results |
|---|---|---|---|---|---|
| Pennycook & Rand 2019 [37] | 1) Bullshit receptivity; 2) Overclaiming knowledge; 3) Analytic thinking; 4) Familiarity of the claim. | M = 1606, MTurk, U.S. | **Study 1:** Cognitive reflection test (CRT) → Pseudo-profound bullshit receptivity task → Random assignment to six headlines presented as Facebook posts → Rating headlines' credibility → Overclaiming questionnaire → Demographic questions. **Study 2:** Random assignment to 10 headlines presented as Facebook posts following a 2 (true/false) x 2 (source/no source) design → outcome measurement → Cognitive reflection test (CTR) → Pseudo-profound bullshit receptivity task → Demographic questions. **Study 3:** Random assignment to 30 headlines presented as Facebook posts following a 2 (true/false) x 3 (pro-Democrat/pro-Republican/neutral—within-subject) design → Outcome measurement → Cognitive reflection test → Pseudo-profound bullshit receptivity task → Demographic questions. | 1) Accuracy of the claim in the headline (4-point scale); 2) Media truth discernment; 3) Sharing intent. | 1) Pseudo-profound bullshit receptivity and the tendency to overclaim were positively correlated with perceived accuracy of fake news and negatively correlated with the tendency to think analytically; 2) Bullshit receptivity was associated with more fake and real news sharing on social media; 3) Familiarity was associated with increased perceived accuracy of headlines, real and fake alike. |
| Jones-Jang et al. 2019 [44] | 1) Media literacy; 2) Information literacy; 3) News literacy; 4) Digital literacy. | N = 1300, national sample of U.S. residents. | Media literacy, information literacy, news literacy and digital literacy measures → Random assignment to 10 political headlines (true/false) → Outcome measurement → Demographic and attitudinal questions. | Identify whether the headline is fake or real (binary). | Identification of fake news was significantly associated with information literacy but not with other literacy types. |
| Fernández-López & Perea 2020 [49] | 1) Language of news (native/foreign); 2) Emotionality. | N = 144, college students, Spanish. | **Study 1:** Random assignment to four fake news items (topics: celebrities/science/ environment); language: (English/Spanish) → Outcome measurement. **Study 2:** Random assignment to one fake news item (language: English/Spanish) → Outcome measurement → Emotionality measurement. | Credibility of a news story (10-point scale) | 1) No difference between credibility of fake news in foreign and native language; 2) Increased negative emotionality was associated with higher fake news credibility. |
| Kim, Moravec & Dennis 2019 [50] | 1) Expert rating: an aggregate of fact-checkers' ratings of the source's articles; 2) User rating: an aggregate of user ratings of the source's articles; 3) User source rating (users directly rate the sources); 4) Headline's belief consistency. | N1 = 590, Qualtrics panel, U.S.; N2 = 299, Qualtrics panel, U.S. | **Study 1:** Explanation of rating system → Random assignment to 8 ideologically counterbalanced headlines* presented as Facebook posts (topic: abortions) following a 3 (rating type: expert rating/user article rating/user source rating) x 3 (rating: low/medium/high—within-subject) design → Outcome measurement. *2 out of 8 headlines were presented in a control condition (without rating) to each participant before the rating explanation. **Study 2:** The second study replicates the first with minor changes in the order of exposure to questionnaire and stimulus blocks. | 1) Believability of the headline; 2) Likelihood of reading, liking, commenting on, and sharing the article. | 1) Ratings affected believability only when sources were rated low; articles from high-rated sources were not more believable; 2) Articles from low-rated sources were less believable when the ratings were from experts or users evaluating articles (and not sources directly); 3) Headline belief consistency increased both perceived believability and engagement intent. |
| Clayton et al. 2019b [51] | 1) General warning about misleading articles; 2) Facebook-style "Disputed" tag under false headlines; 3) "Rated false" tag under false headlines; 4) Headline political concordance. | N = 2994, MTurk, U.S. | Demographic and attitudinal questions → Rating the accuracy of several real and fabricated political statements → Political knowledge questions → Random assignment to nine political headlines presented as Facebook posts following a 2 (warning/no warning) x 3 (tag: none/disputed/rated false) x 2 (true/false) → Outcome measurement. *The study also included a pure control group that was not exposed to the headlines. | 1) Accuracy of the claim in the headline (4-point scale); 2) Liking and sharing intent. | 1) Both "Disputed" and "Rated false" tags reduced belief in false news; "Disputed" tag: 10% reduction in the proportion of respondents who rated false headlines as credible; "Rated false" tag: 13% reduction. 2) These effects do not vary by political concordance of headlines; 3) General warnings have a small effect on perceived accuracy of false news, while also reducing belief in real news; 4) Warnings had no effect on likelihood of liking or sharing the news. |
| Lutzke et al. 2019 [36] | Critical thinking prime, two variations. | N = 2750, Qualtrics panel, U.S. | Random assignment to one of the conditions following a 2 (true/false) x 3 (intervention type: reading guidelines for evaluating news online/reading same guidelines and rating importance of each one/control—no intervention) design → Exposure to a news story about climate change presented as a Facebook post → Outcome measurement → Measurement of knowledge about climate change → Demographic questions. | 1) Credibility scale (trustworthiness and accuracy), 10-point scale; 2) Liking and sharing intent. | Both types of exposure to guidelines produced small but significant reduction in the likelihood to trust, like, or share fake but not real news. |
| Guess et al. 2020 [52] | 1) Exposure to a set of media literacy guidelines; 2) Headline political concordance. | N1 = 4907, YouGov panel, U.S.; N2 = 3200, MTurk and Online Bureau survey panel, India, online; N3 = 3700, India, face-to-face. | **U.S. study** Random assignment to either exposure to media literacy guidelines ('Tips to spot false news'), or placebo → Exposure to eight out of 16 (1st wave) /all 16 (2nd wave) news headlines presented as Facebook posts, counterbalanced by ideological valence, source prominence, and veracity → Outcome measures. **India study** Random assignment to either exposure to media literacy guidelines, or placebo → Respondents in exposure condition shown (online) or read (face-to-face) six tips for spotting false news → Exposure to 12 headlines in text format (online survey) or 16 headlines read out by enumerators (face-to-face survey), counterbalanced by veracity and political stance → Outcome measurement. | 1) Accuracy of the claim in the headline (4-point scale); 2) Discernment (calculated at the respondent level as the mean difference in perceived accuracy between all false and all mainstream news headlines viewed); 3) Sharing intent. | 1) Intervention reduced the perceived accuracy of both true and false headlines, with larger effects on false headlines. Discernment rates thus improved in the U.S. and India online samples (but not in the rural offline sample in India). The effect was not moderated by partisan congeniality; 2) The effect was identifiable after several weeks in the U.S., but not in India; 3) The intervention increased sharing intent for real news and decreased it for hyperpartisan news. |

*(Continued)*

**Table 1.** (Continued)

| Author(s), year | Factors | Sample | Procedures and stimuli* | Outcome variables & measurement | Results |
|---|---|---|---|---|---|
| Pennycook et al. 2020 [34] | 1) Accuracy judgement task vs sharing task; 2) Accuracy induction intervention. | N = 2000, quota sample via Lucid, U.S. | **Study 1:** Random assignment to either rating accuracy task or rating sharing intent task → Exposure to 30 news headlines related to COVID-19, presented as Facebook posts, either true or false → Outcome measurement. **Study 2:** Random assignment to either 1) Accuracy induction: judging the accuracy of a nonCOVID-19-related headline) or 2) No accuracy induction → Exposure to 30 news headlines related to COVID-19, presented as Facebook posts, either true or false → Outcome measurement. | 1) Accuracy of the claim in the headline (binary); 2) Discernment (difference between responses to true headlines and false headlines); 3) Sharing intent (binary in study 1, 6-point scale in study 2). 4) Sharing discernment: reported intention to share true vs false headlines. | 1) Discernment was higher for accuracy judgments compared to sharing intentions; 2) People in the accuracy nudge condition were significantly more likely to share true stories compared to false ones. |
| Morris et al. 2020 [53] | 1) Variations of fact-checking inoculation; 2) Political concordance. | N = 1284, MTurk, U.S. | Random assignment to one of the prompts prior to stimulus exposure: 1) All items deemed incorrect by non-partisan fact-checkers (Inoculation); 2) Some are correct & some deemed incorrect by fact-checkers; 3) Some are disputed by some members of the media; 4) Control → Exposure to 12 fake political news stories (ideologically counterbalanced) → Outcome measurement → Attitudinal and demographic questions. | Believability of news stories (5-point scale). | Fact-checking inoculation did have a visible, but marginally insignificant effect (decrease with p = 0.55 in the main model) on believability of fake news, however this effect is largely driven by liberal respondents. |
| Garrett & Poulsen 2019 [54] | 1) Type of Facebook flag; 2) Headline political concordance. | N1 = 226, Federated Sample; N2 = 858, Survey Sampling International, both U.S | **Study 1:** Wave 1: Participants are asked to sign up in their Facebook accounts → Attitudinal and demographic questions. Wave 2: Participants recontacted in two weeks → Random assignment to two popular political falsehoods presented as Facebook posts, ideologically counterbalanced, with one of the flag types: Peer-generated/Fact-checker/Self-identified humor/Control) → Outcome measurement. **Study 2:** Random assignment to exposure to two popular political falsehoods presented as Facebook post, ideologically counterbalanceв, with one of humor flag types:Story self-identified as humor/Story designated by Facebook as humor/Publisher self-identified/Publisher designated by Facebook/Control → Distractor task → Outcome measurement → Manipulation check. | 1) Acceptance / perceived accuracy of the false claim in the headline (2-item 7-point scale, avg) 2) Sharing intent (8-item 7-point scale); 3) Source credibility (8-item 7-point scale, avg). | Self-identified humor flagging decreased perceived accuracy, sharing intent, and perceived credibility of the source by 11–20% of the entire scale in some models, regardless of political consistency; other types of flags were ineffective. |
| Van Duyn & Collier 2018 [26] | 1) Fake news priming by elites. | N = 299, Mturk, U.S. | Random assignment to one of two sets of nine tweets from politicians (topic: fake news/federal budget) → Manipulation check → Randomized exposure to one of two sets of three articles (all true/all false) → Outcome measurement. | 1) Whether an article was real or fake (3-point scale); 2) Media trust. | 1) Participants primed with elite fake news discourse identified real news with less accuracy; 2) Those primed with elite fake news discourse reported lower overall media trust. |
| Tsang 2020 [55] | 1) Source (Legacy news outlet / No Source / Online forum) 2) Policy support | N = 50 (pilot study), N = 280 (main experiment), recruited via Dynata | Attitudinal questions → Random assignment to a fake news story mimicking a WhatsApp post in one of three conditions (Source: legacy News outlet/no source/online forum) → Outcome measurement. | 1) Perceived news fakeness: scales (1–5) indicating whether the news was (a) invented, (b) fabricated, and (c) could be considered fake news; 2) Perceived inaccuracy: scales (1–5) measuring whether the news was perceived as (a) misleading, (b) contained exaggeration, and (c) involved serious errors; Perceived intent (political motivation) scale (1–5). | 1) News source did not have an effect on the perception of news fakeness, inaccuracy or intent. 2) Motivated fake news perception (policy support) had an impact on news fakeness, perceived inaccuracy, and perceived intent. |
| Roozenbeek & van der Linden, 2019 [56] | Inoculation intervention in the form of an online game where players learn about various misinformation strategies | Online sample of N = 15000, recruited via a university press release | Introduction to a game → Approx. 15 min. of playtime, choice-based architecture where players earn badges by applying six popular misinformation techniques to hypothetical situations → Assessing credibility of 6 headlines and tweets pre- and post-game play (2 control, not misinformation; 4 fake, randomized, each using one of the techniques embedded in the game). | Message reliability scale (1–7), measured before and after playing as part of gameplay. | 1) Active inoculation conferred by playing the game significantly reduced the perceived reliability of misinformation. 2) Significant difference in pre-scores and post-scores was detected for fake tweets and headlines, but not for real headlines. 3) Differences in inoculation effects across genders, education levels, age groups, or political ideologies can be considered negligible. |

*(Continued)*

**Table 1.** (Continued)

| Author(s), year | Factors | Sample | Procedures and stimuli* | Outcome variables & measurement | Results |
|---|---|---|---|---|---|
| Amazeen & Bucy, 2019 [57] | Procedural news knowledge (The PNK measure included 10 multiple choice questions, each with one correct answer) | N1 = 770, recruited by Survey Sampling International; N2 = 1,067, ProdegeMR market research panel. | Sample 1: Attitudinal questions; News knowledge assessment → Some participants were primed with a message about media literacy → Questions about native advertising → Random assignment to one of two native ads (political/non-political) → Distractor task → Outcome measurement → Demographic questions. Sample 2: Demographic and attitudinal questions → News knowledge assessment → Random assignment to 10 political headlines (true/fake) → Outcome measurement. | 1) Recognition of native advertising (one closed-ended and two open-ended questions); 2) Perceived accuracy of news headlines (4-point scale); 3) Perceived threat (six 7-point scales with bipolar adjective pairs); 4) Counterarguing (open-ended responses coded by the researchers); 5) Persuasion: intent to "like" and "share" (for all participants) or to purchase (only for those exposed to native ads). | 1) Higher level of procedural news knowledge (PNK) increased the odds of native ad recognition; 2) The level of PNK was negatively associated with the perceived accuracy of fabricated news; 3) PNK was positively associated with the perceived threat of being confronted with native advertising; 4) Participants with greater levels of PNK were more likely to counterargue when viewing native advertising; 5) Participants with lower levels of PNK found the native ad more persuasive than those with higher levels. |

* Note: In study design statements, all factors are between-subjects unless stated otherwise.

Having extracted and classified the dependent variables in the reviewed studies, we proceed to mapping our observations against the factors and correlates that were theorized to exert effects on them (see Table 3).

We observed that the experimental studies in our sample measure or manipulate three types of factors hypothesized to influence individuals' belief in fake news. The first category encompasses variables related to the news message, the way it is presented, or the features of the information environment where exposure to information occurs. In other words, these tests seek to answer the question: What kinds of fake news are people more likely to fall for? The second category takes a different approach and examines respondents' individual traits predictive of their susceptibility to disinformation. Put simply, these tests address the broad question of who falls for fake news. Finally, the effects of measures specifically designed to combat the spread of fake news constitute a qualitatively distinct group. Granted, this is a necessarily simplified categorization, as factors do not always easily lend themselves to inclusion into one of these baskets. For example, the effect of a pro-attitudinal message can be seen as a combination of both message-level (e. g. conservative-friendly wording of the headline) and an individual-level predisposition (recipient embracing politically conservative views). For presentation purposes, we base our narrative synthesis of the reviewed evidence on the following categorization: 1) Factors residing entirely outside of the individual recipient (message features, presentation cues, information environment); 2) Recipient's individual features; 3) Interventions. For each category, we discuss theoretical frameworks that the authors employ and specific study designs.

## Findings

A fundamental question at the core of many investigations that we reviewed is whether people are generally predisposed to believe fake news that they encounter online. Previous research suggests that individuals go about evaluating the veracity of falsehoods similarly to how they

**Table 2. Description of outcome variables.**

| Outcome | Description | # of observations |
|---|---|---|
| Perceived credibility of the message | Respondent's perception of credibility, believability or accuracy of the message/headline. Can be measured for both real and fake news items, commonly on a Likert-type scale. Some studies adapt existing multi-item credibility scales. | 22 |
| Sharing intent | Self-reported likelihood to share the headline/message on social media | 13 |
| Discernment/Detection accuracy | Accuracy of a credibility judgement made by a respondent relative to the "ground truth" of the message/headline. Used in the studies that expose participants to both fake and real news to construct respondent-level discernment scores. In most instances derived from credibility scores. | 10 |
| Engagement | Self-reported likelihood to read, like, post a comment, etc. (but not share). | 5 |
| Source credibility | Perceived credibility of a source of the message/headline. | 2 |
| Influence on factual beliefs | Belief in a false statement contained in a message. | 2 |
| Sharing discernment | The difference in the reported sharing likelihood of true relative to false messages/headlines. | 1 |
| Media trust | General trust in news media following the exposure to experimental treatment. | 1 |
| Perceived inaccuracy | To what extent the message is perceived as misleading, exaggerated, or involving errors. | 1 |
| Perceived political intent | Perception that author or source has political motives, such as supporting a political candidate, influencing votes, or swaying public opinion. | 1 |
| Recognition of native advertising | Ability to distinguish editorial news content from commercial content. | 1 |
| Perceived threat | Perceived threat of being confronted with a covert persuasive attempt. | 1 |

Note: A single study could yield several observations if it considered multiple outcome variables.

process true information [38]. Generally, most individuals tend to accept information that others communicate to them as accurate, provided that there are no salient markers suggesting otherwise [39].

Informed by these established notions, some of the authors whose work we reviewed expect to find the effects of "truth bias," a tendency to accept all incoming claims at face value, including false ones. This, however, does not seem to be the case. No study under review reported the majority of respondents trusting most fake messages or perceiving false and real messages as equally credible. If anything, in some cases a "deception bias" emerges, where individuals' credibility judgements are biased in the direction of rating both real and false news as fake. For example, Luo et al. [40] found that across two experiments where stimuli consisted of equal numbers of real and fake headlines participants were more likely to rate all headlines as fake, resulting in just 44.6% and 40% of headlines marked as real across two studies. Yet, it is possible that this effect is a product of the experimental setting where individuals are alerted to the possibility that some of the news is fake and prompted to scrutinize each message more thoroughly than they would while leisurely browsing their newsfeed at home.

The reviewed evidence of individuals' overall credibility perceptions of fake news as compared to real news, as well as of people's ability to tell one from another, is somewhat contradictory. Several studies that examined participants' accuracy in discerning real from fake news report estimates that are either below or indistinguishable from random chance: Moravec et al. [41] report a mean detection rate of 43.9%, with only 17% of participants performing better

**Table 3. Number of observations for each factor/correlate and the outcome type.**

| Factor group | Factor | Perceived credibility of a message | Sharing intent | Discernment/Detection accuracy | Engagement | Source credibility | Overall |
|---|---|---|---|---|---|---|---|
| *Message-level* | Trusted endorsements on social media | 3 | 1 | | 1 | | 5 |
| | Bandwagon cues on social media | 4 | 2 | | 1 | | 7 |
| | Truth-bias | 1 | | 1 | | | 2 |
| | News topic | 1 | | 1 | | | 2 |
| | Negative comments on social media | 1 | 1 | | | | 2 |
| | Message content and format cues | 1 | 1 | | | 1 | 3 |
| | Topic's personal relevance | 1 | 1 | | | 1 | 3 |
| | Self-reported information consumption habits | 1 | 1 | | | 1 | 3 |
| | Source reliability ratings | 2 | 2 | | 1 | 1 | 6 |
| | Presentation format: highlighting source over headline | 1 | 1 | | 1 | | 3 |
| | Political concordance/belief consistency | 11 | 7 | 3 | 3 | 1 | 25 |
| *Individual-level* | Need for cognition | 1 | 1 | | | 1 | 3 |
| | Faith in intuition | 1 | 1 | | | 1 | 3 |
| | Processing fluency via prior exposure/familiarity | 2 | 2 | 1 | | | 5 |
| | Propensity to engage in analytical reasoning | 3 | 2 | 3 | | | 8 |
| | Conditions favoring heightened deliberation | 1 | | 1 | | | 2 |
| | Delusion-like ideation | 1 | | 1 | | | 2 |
| | Actively open-minded thinking | 1 | | 1 | | | 2 |
| | Dogmatism | 1 | | 1 | | | 2 |
| | Religious fundamentalism | 1 | | 1 | | | 2 |
| | Bullshit receptivity | 1 | 1 | 1 | | | 3 |
| | Overclaiming | 1 | 1 | 1 | | | 3 |
| | Emotionality/emotional processing | 2 | | 1 | | | 3 |
| | Literacy (various types) | 1 | 1 | 1 | | | 3 |
| *Interventions & ecological factors* | Critical thinking/accuracy prime | 2 | 2 | 1 | 1 | | 6 |
| | Exposure to media literacy guidelines (incl. game format) | 2 | 1 | 1 | | | 4 |
| | General fact-checking/ misinformation warning | 2 | 1 | | 1 | | 4 |
| | Social media flags | 4 | 3 | | 1 | 1 | 9 |
| **Overall** | | 54 | 32 | 20 | 10 | 8 | |

Note: Only outcome variables with more than one observation are included in the table.

A single study could yield several observations if it considered multiple independent and/or outcome variables.

than chance; in Luo et al. [40], detection accuracy is slightly better than chance (53.5%) in study 1 and statistically indistinguishable from chance in study 2 (49.2%). Encouragingly, the majority of other studies where respondents were exposed to both real and fake news items provide evidence suggesting that people's average capacity to tell one from another is considerably greater than chance. In all studies reported in Pennycook and Rand [25], average perceived credibility of real headlines is above 2.5 on a four-point scale from 1 to 4, while average credibility of fake headlines is below 1.6. A similar distance—about one point on a four-point

scale—marks the difference between real and fake news' perceived credibility in experiments reported in Bronstein et al. [42]. In Bago et al. [43], participants rated less than 40% of fake headlines and more than 60% of real headlines as accurate. In Jones-Jang et al. [44], respondents correctly identified fake news 6.35 attempts out of 10.

Following the aggregate-level assessment, we proceed to describing three main groups of factors that researchers identify as sources of variation in perceived credibility of fake news.

## Message-level and environmental factors

When apparent signs of authenticity or fakeness of a news item are not immediately available, individuals can rely on certain message characteristics when making a credibility judgement. Two major message-level factors stand out in this cluster of evidence as most frequently tested (see Table 3). Firstly, alignment of the message source, topic, or content with the respondent's prior beliefs and ideological predispositions; secondly, social endorsement cues. Theoretical expectations within this approach are largely shaped by dual-process models of learning and information processing [58, 59] borrowed from the field of psychology and adapted for online information environments. These theories emphasize how people's information processing can occur through either the more conscious, analytic route or the intuitive, heuristic route. The general assumption traceable in nearly every theoretical argument is that consumers of digital news routinely face information overload and have to resort to fast and economical heuristic modes of processing [60], which leads to reliance on cues embedded in messages or the way they are presented. For example, some studies that examine the influence of online social heuristics on evaluations of fake news' credibility build on Sundar's [61] concept of bandwagon cues, or indicators of collective endorsement of online content as a sign of its quality. More generally, these studies continue the line of research investigating how perceived social consensus on certain issues, gauged from online information environments, contributes to opinion formation (e. g. Lewandowsky et al. [62]).

Exploring the interaction between message topic and bandwagon heuristics on perceived credibility of fake news headlines, Luo et al. [40] find that a high number of likes associated with the post modestly increases (by 0.34 points on a 7-point scale) perceived credibility of both real and fake news compared to few likes. Notably, this effect is observed for health and science headlines, but not for political ones. In contrast, Kluck et al. [35] fail to find the effect of the numeric indicator of Facebook post endorsement on perceived credibility. This discrepancy could be explained by differences in the design of these two studies: whereas in Luo et al. participants were exposed to multiple headlines, both real and fake, Kluck et al. only assessed perceived credibility of just one made-up news story. This may have led to the unique properties of this single news story contributing to the observed result., Kluck et al. further reveal that negative comments questioning the stimulus post's authenticity do dampen both perceived credibility (by 0.21 standard deviations) and sharing intent. In a rare investigation of news evaluation on Instagram, Mena et al. [46] demonstrate that trusted endorsements by celebrities do increase credibility of a made-up non-political news post, while bandwagon endorsements do not. Again, this study relies on one fabricated news post as a stimulus. These discrepant results of social influence studies suggest that the likelihood of detecting such effects may be contingent on specific study design choices, particularly the format, veracity, and sampling of stimulus messages. Generalizability and comparability of the results generated in experiments that use only one message as a stimulus should be enhanced by replications that employ stimulus sampling techniques [63].

Following one of the most influential paradigms in political communication research—the motivated reasoning account postulating that people are more likely to pursue, consume,

endorse and otherwise favor information that matches their preexisting beliefs or comes from an ideologically aligned source—most studies in our sample measure the ideological or political concordance of the experimental messages and most commonly use it in statistical models as covariates or hypothesized moderators. Where they are reported, the pattern of direct effects of ideological concordance largely conforms to expectations, as people tend to rate congenial messages as more credible. In Bago et al. [43], headline political concordance increased the likelihood of participants rating it as accurate (b = 0.21), which was still meager compared to the positive effect of the headline's actual veracity (b = 1.56). In Kim, Moravec and Dennis [50], headline political concordance was a significant predictor of believability (b = 0.585 in study 1; b = 0.153 in study 2), but the magnitude of this effect was surpassed by that of low source ratings by experts (b = −0.784 in study 1; b = -0.365 in study 2). In turn, increased believability heightened the reported intent to read, like, and share the story. In the same study, both expert and user ratings of the source displayed alongside the message influenced its perceived believability in both directions. According to the results of the study by Kim and Dennis [14], increased relevance and pro-attitudinal directionality of the statement contained in the headline predicted increased believability and sharing intent. Similarly, Moravec et al. [41] argued that the confirmatory nature of the headline is the single most powerful predictor of belief in false but not true news headlines. Tsang [55] found sizable effects of the respondents' stance on the Hong Kong extradition bill on perceived fakeness of a news story covering the topic in line with the motivated reasoning mechanism.

At the same time, the expectation that individuals will use the ideological leaning of the source as a credibility cue when faced with ambiguous messages lacking other credibility indicators was not supported by data. Relying on the data collected from almost 4000 Amazon Mechanical Turk workers, Clayton et al. [45] failed to detect the hypothesized influence of motivated reasoning, induced by the right or left-leaning mainstream news source label, on belief in a false statement presented in a news report.

Several studies tested the effects of factors beyond social endorsement and directional cues. Schaewitz et al. [13] looked at the effects of such message characteristics as source credibility, content inconsistencies, subjectivity, sensationalism, and the presence of manipulated images on message and source credibility appraisals, and found no association between these factors and focal outcome variables—against the background of the significant influence of personal-level factors such as the need for cognition. As already mentioned, Luo et al. [40] found that fake news detection accuracy can also vary by the topic, with respondents recording the highest accuracy rates in the context of political news—a finding that could be explained by users' greater familiarity and knowledge of politics compared to science and health.

One study under review investigated the possibility that news credibility perceptions can be influenced not by the features of specific messages, but by characteristics of a broader information environment, for example, the prevalence of certain types of discourse. Testing the effects of exposure to the widespread elite rhetoric about "fake news," van Duyn and Collier [26] discovered evidence that it can dampen believability of *all* news, damaging people's ability to identify legitimate content in addition to reducing general media trust. These effects were sizable, with primed participants ascribing real articles on average 0.47 credibility points less than those who haven't been exposed to politicians' tweets about fake news, on a 3-point scale.

As this brief overview demonstrates, the message-level approaches to fake news susceptibility consider a patchwork of diverse factors, whose effects may vary depending on the measurement instruments, context, and operationalization of independent and outcome variables. Compared to individual-level factors, scholars espousing this paradigm tend to rely on more diverse experimental stimuli. In addition to headlines, they often employ story leads and full news reports, while the stimulus new stories cover a broader range of topics than just politics.

At the same time, out of ten studies attributed to this category, five used either one or two variations of a single stimulus news post. This constitutes an apparent limitation to the generalizability of their findings. To generate evidence generalizable beyond specific messages and topics, future studies in this domain should rely on more diverse sets of stimuli.

## Individual-level factors

This strain of research recognizes the differences in people's individual cognitive styles, predispositions, and conditions as the main source of variation in fake news credibility judgements. Theoretically, they largely rely on dual-process approaches to human cognition as well [64, 65]. Scholars embracing this approach explain some people's tendency to fall for fake news by their reliance, either innate or momentary, on less analytical and more reflexive modes of thinking [37, 42]. Generally, they tend to ascribe fake news susceptibility to lack of reasoning rather than to directionally motivated reasoning.

Pennycook and Rand [25] employ the established measure of analytical thinking, the Cognitive Reflection Test, to demonstrate that respondents who are more prone to override intuitive thinking with further reflection are also better at discerning false from real news. This effect holds regardless of whether the headlines are ideologically concordant or discordant with individuals' views. Importantly, the authors also find that headline plausibility (understood as the extent to which it contains a statement that sounds outrageous or patently false to an average person) moderates the observed effect, suggesting that more analytical individuals can use extreme implausibility as a cue indicating news' fakeness.

In a 2020 study [37], Pennycook and Rand replicated the relationship between CRT and fake news discernment, in addition to testing novel measures—pseudo-profound bullshit receptivity (the tendency to ascribe profound meaning to randomly generated phrases) and a tendency to overclaim one's level of knowledge—as potential correlates of respondents' likelihood to accept claims contained in false headlines. Pearson's r ranged from 0.30 to 0.39 in study 1 and from 0.20 to 0.26 in study 2 (all significant at p<0.001 in both studies), indicating modestly sized yet significant correlations. All three measures were correlated with perceived accuracy of fake news headlines as well as with each other, based on which the authors speculated that these measures are all connected to a common underlying trait that manifests as the propensity to uncritically accept various claims of low epistemic value. The researchers labelled this trait *reflexive open-mindedness*, as opposed to *reflective open-mindedness* observed in more analytical individuals. In a similar vein, Bronstein et al. [42] added cognitive tendencies such as delusion-like ideation, dogmatism, and religious fundamentalism to the list of individual-level traits weakly associated with heightened belief in fake news, while analytical and open-minded thinking slightly decreased this belief.

Schaewitz et al. [13] linked the classic concept from credibility research, need for cognition, to the tendency to rate down credibility (in some models but not others) and accuracy of non-political fake news. This concept overlaps with analytical thinking from Pennycook and Rand's experiments, yet distinct in that it captures the self-reported pleasure from (and not just the proneness to) performing cognitively effortful tasks.

Much like the studies reviewed above, experiments by Martel et al. [48] and Bago et al. [43] challenged the motivated reasoning argument as applied to fake news detection, focusing instead on the classical reasoning explanation: the more analytic the reasoning, the higher the likelihood to accurately detect false headlines. In contrast to the above accounts, both studies investigate the momentary conditions, rather than stable cognitive features, as sources of variation in fake news detection accuracy. In Martel et al. [48], increased emotionality (as both the current mental state at the time of task completion and the induced mode of information

processing) was strongly associated with the increased belief in fake news, with induced emotional processing resulting in a 10% increase in believability of false headlines. Fernández-López and Perea [49] reached similar conclusions about the role of emotion drawing on a sample of Spanish residents.

Bago et al. [43] relied on the two-response approach to test the effects of the increased time for deliberation on perceived accuracy of real and false headlines. Compared to the first response, given under time constraints and additional cognitive load, the final response to the same news items for which participants had no time limit and no additional cognitive task indicated significantly lower perceived accuracy of fake (but not real) headlines, both ideologically concordant and discordant. The effect of heightened deliberation (b = 0.36) was larger than the effect of headline political concordance (b = -0.21). These findings lend additional support to the argument that decision conditions favoring more measured, analytical modes of cognitive processing are also more likely to yield higher rates of fake news discernment.

Pennycook et al. [47] provide evidence supporting the existence of the illusory truth effect—the increased likelihood to view the already seen statements as true, regardless of the actual veracity—in the context of fake news. In their experiments, a single exposure to either a fake or real news headline slightly yet consistently (by 0.09 or 0.11 points on a 4-point scale) increased the likelihood to rate it as true on the second encounter, regardless of political concordance, and this effect persists after as long as a week.

It is not always how individuals process messages, but how competent they are about the information environment, that affects their ability to resist misinformation. Amazeen and Bucy [57] introduce a measure of procedural news knowledge (PNK), or working knowledge of how news media organizations operate, as a predictor of the ability to identify fake news and other online messages that can be viewed as deliberately deceptive (such as native advertising). In their analysis, one standard deviation decrease in PNK increased perceived accuracy of fabricated news headlines by 0.19 standard deviation. Interestingly, Jones-Jang et al. [44] find a significant correlation between information literacy (but not media and news literacies) and identification between fake news stories.

Taken together, the evidence reviewed in this section provides robust support to the idea that analytic processing is associated with more accurate discernment of fake news. Yet, it has to be noted that the generalizability of these findings could be constrained by the stimulus selection strategy that many of these studies share. All experiments reviewed above, excluding Schaewitz et al. [13] and Fernández-López and Perea [49], rely on stimulus material constructed from equal shares of real mainstream news headlines and real fake news headlines sourced from fact-checking websites like *Snopes.com*. As these statements are intensely political and often blatantly untrue, the sheer implausibility of some of the headlines can offer a "fakeness" cue easily picked up by more analytical—or simply politically knowledgeable—individuals, a proposition tested by Pennycook and Rand [25]. While they preserve the authenticity of the information environment around the 2016 U.S. presidential election, it is unclear what these findings can tell us about the reasons behind people's belief in fake news that are less egregiously "fake" and therefore do not carry a conspicuous mark of falsehood.

## Accuracy-promoting interventions

The normative foundation of much of the research investigating the reasons behind people's vulnerability to misinformation is the need to develop measures limiting its negative effects on individuals and society. Two major approaches to countering fake news and its negative effects can be distinguished in the literature under review. The first approach, often labelled inoculation, is aimed at preemptively alerting individuals to the dangers of online deception and

equipping them with the tools to combat it [44, 56]. The second manifests in tackling specific questionable news stories or sources by labelling them in a way that triggers increased scrutiny by information consumers [51, 54]. The key difference between the two is that inoculation-based strategies are designed to work preemptively, while labels and flags are most commonly presented to information consumers alongside the message itself.

Some of the most promising inoculation interventions are those designed to enhance various aspects of media and information literacy. Recent studies demonstrated that preventive techniques—like exposing people to anti-conspiracy arguments [66] or explaining deception strategies [67]—can help neutralize harmful effects of misinformation before the exposure. Grounded in the idea that the lack of adequate knowledge and skills among news consumers makes people less critical and, thus, more susceptible to fake news [68], such measures aim at making online deception-related considerations salient in the minds of large swaths of users, as well as at equipping them with basic techniques that help spot false news.

In a cross-national study that involved respondents from the United States and India, Guess et al [52] find that exposing users to a set of simple guidelines for detecting misinformation modelled after similar Facebook guidelines (e.g., "Be skeptical of headlines," "Watch for unusual formatting") improves fake news discernment rate by 26% in the U.S. sample and by 19% in the Indian sample, regardless of whether the headlines are politically concordant or discordant. These effects persist several weeks post-exposure. Interestingly, it might be that the effect is caused not so much by participants heeding the instructions as by simply priming them to think about accuracy. When testing the effects of accuracy priming in the context of COVID-19 misinformation, Pennycook et al. [34] reveal that inattention to accuracy considerations is rampant: people asked whether they would share false stories appear to rarely consider their veracity unless prompted to do so. Yet, asking them to rate the accuracy of a single unrelated headline before going into the task dramatically improved accuracy and reduced the likelihood to share false stories: the difference in sharing likelihood of true relative to false headlines was 2.8 times higher in the treatment group comparatively to the control group.

On a more general note, the latter finding could suggest that the results of all experiments that include false news discernment tasks could be biased in the direction of more accuracy simply by the virtue of priming participants to think about news' veracity, compared to their usual state of mind when browsing online news. Lutzke et al. [36] reach similar results when they prime critical thinking in the context of climate change news, resulting in diminished trust and sharing intentions for falsehoods even among climate change doubters.

A study by Roozenbeek and van der Linden [56] demonstrated the capacity of a scalable inoculation intervention in the format of a choice-based online game to confer resistance against several common misinformation strategies. Over the average of 15 minutes of gameplay, users were tasked with choosing the most efficient ways of misinforming the audience in a series of hypothetical scenarios. Post-gameplay credibility scores of fake news items embedded in the game were significantly lower than pre-test scores using a one-way repeated measures $F(5, 13559) = 980.65$, Wilk's $\Lambda = 0.73$, $p < 0.001$, $\eta^2 = 0.27$. These findings were replicated in a between-subjects design with a control group in Basol et al. [69], although this study was not included in our sample based on formal criteria.

Fact-checking is arguably the most publicly visible format of real measures used to combat online misinformation. Studies in our sample present mixed evidence of the effectiveness of fact-checking interventions in reducing credibility of misinformation. Using different formats of fact-checking warnings before exposing participants to a set of verifiably fake news stories, Morris et al. [53] demonstrated that the effects of such measures can be limited and contingent on respondents' ideology (liberals tend to be more responsive to fact-checking warnings than conservatives). Encouragingly, Clayton et al. [51] found that labels indicating the fact that a

particular false story has been either disputed or rated false do decrease belief in this story, regardless of partisanship. The "Disputed" tag placed next to the story headline decreased believability by 10%, while the "Rated false" tag was 13% effective. At the same time, in line with van Duyn and Collier [26], they showed that general warnings that are not specific to particular messages are less effective and can reduce belief in real news. Finally, Garrett and Poulsen [54], comparing the effects of three types of Facebook flags (fact-checking warning; peer warning; humorous label) found that only self-identification of the source as humorous reduces both belief and sharing intent. The discrepant conclusions that these three studies reach are unsurprising given differences in format and meaning of warnings that they test.

In sum, findings in this section suggest that the general warnings and non-specific rhetoric of "fake news" should be employed with caution so as to avoid the outcomes that can be opposite to the desired effects. Recent advances in scholarship on the backfire effect of misinformation corrections have called into question the empirical soundness of this phenomenon [70, 71]. However, multiple earlier studies across several issue contexts have documented specific instances where attitude-challenging corrections were linked to compounding misperceptions rather than rectifying them [72, 73]. Designers of accuracy-promoting interventions should at least be aware of the possibility that such effects could follow.

Overall, while the evidence of the effects of labelling and flagging specific social media messages and sources remains inconclusive, it appears that priming users to think of online news' accuracy is a scalable and cheap way to improve the rates of fake news detection. Gamified inoculation strategies also hold potential to reach mass audiences while preemptively familiarizing users with the threat of online deception.

## Discussion

We have applied a scoping review methodology to map the existing evidence of the effects various antecedents to people's belief in false news, predominantly in the context of social media. The research landscape presents a complex picture, suggesting that the focal phenomenon is driven by the interplay of cognitive, psychological and environmental factors, as well as characteristics of a specific message.

Overall, the evidence under review speaks to the fact that people on average are not entirely gullible, and they can detect deceitful messages reasonably well. While there has been no evidence to support the notion of "truth bias," i.e., people's propensity to accept most incoming messages as true, the results of some studies in our sample suggested that under certain conditions the opposite—a scenario that can be labelled "deception bias"—can be at work. This is consistent with some recent theoretical and empirical accounts suggesting that a large share of online information consumers today approach news content with skepticism [74, 75]. In this regard, the problem with fake news could be not only that people fall for it, but also that it erodes trust in legitimate news.

At the same time, given the scarcity of attention and cognitive resources, individuals often rely on simple rules of thumb to make efficient credibility judgements. Depending on many contextual variables, such heuristics can be triggered by bandwagon and celebrity endorsements, topic relevance, or presentation format. In many cases, messages' concordance with prior beliefs remains a predictor of increased credibility perceptions.

There is also consistent evidence supporting the notion that certain cognitive styles and predilections are associated with the ability to discern real from fake headlines. The overarching concept of reflexive open-mindedness captures an array of related constructs that are predictive of propensity to accept claims of questionable epistemological value, an entity of which fake news is representative. Yet, while many of the studies focusing on individual-level factors

demonstrate that the effects of cognitive styles and mental states are robust across both politically concordant and discordant headlines, the overall effects of belief consistency remain powerful. For example, in Pennycook and Rand [25] politically concordant items were rated as significantly more accurate than politically discordant items overall (this analysis was used as a manipulation check). This suggests that individuals may not be necessarily engaging in motivated reasoning, yet still using belief consistency as a credibility cue.

The line of research concerned with accuracy-improving interventions reveals limited efficiency of general warnings and Facebook-style tags. Available evidence suggests that simple inoculation interventions embedded in news interfaces to prime critical thinking and exposure to news literacy guidelines can induce more reliable improvements while avoiding normatively undesirable effects.

## Conclusions and future research

The review highlighted a number of blind spots in the existing experimental research on fake news perceptions. Since this literature has to a large extent emerged as a response to particular societal developments, the scope of investigations and study design choices bear many contextual similarities. The sample is heavily skewed toward the U.S. news and news consumers, with the majority of studies using a limited set of politically charged falsehoods for stimulus material. While this approach enhances external validity of studies, it also limits the universe of experimental fake news to a rather narrow subset of this sprawling genre. Future studies should transcend the boundaries of the "fake news canon" and look beyond Snopes and Politifact for stimulus material in order to investigate the effects of already established factors on perceived credibility of misinformation that is not political or has not yet been debunked by major fact-checking organizations.

Similarly, the overwhelming majority of experiments under review seek to replicate the environment where many information consumers encountered fake news during and after the misinformation crisis of 2016, to which end they present stimulus news items in the format of Facebook posts. As a result, there is currently a paucity of studies looking at all other rapidly emerging venues for political speech and fake news propagation: Instagram, messenger services like WhatsApp, and video platforms like YouTube and TikTok.

The comparative aspect of fake news perceptions, too, is conspicuously understudied. The only truly comparative study in our sample [52] uncovered meaningful differences in effect sizes and decay time between U.S. and Indian samples. More comparative research is needed to specify whether the determinants of fake news credibility are robust across various national political and media systems.

Two methodological concerns also stand out. Firstly, a dominant approach to constructing experimental stimuli rests on the assumption that the bulk of news consumption on social media occurs on the level of headline exposure—i.e. users process news and make sharing decisions based largely on news headlines. While there are strong reasons to believe that it is true for some news consumers, others might engage with news content more thoroughly, which can yield differences in effects observed on the headline level. Future studies could benefit from accounting for this potential divergence. For example, researchers can borrow the logic of Arceneaux and Johnson [76] and introduce an element of choice, thus enabling comparisons between those who only skim headlines and those who prefer to click on articles to read.

Finally, the results of most existing fake news studies could be systematically biased by the mere presence of a credibility assessment task. As Kim and Dennis [14] argue, browsing social media feeds is normally associated with a hedonic mindset, which is less conducive to critical

assessment of information compared to a utilitarian mindset. This is corroborated by Penny-cook et al. [34] who show that people who are not primed to think about accuracy are significantly more likely to share false news. A small credibility rating task produces massive accuracy improvement, underscoring the difference that a simple priming intervention can make. Asking respondents to rate credibility of treatment news items could work similarly, thus distorting the estimates compared to respondents' "real" accuracy rates. In this light, future research should incorporate indirect measures of perceived fake and real news accuracy that could measure the focal construct without priming respondents to think about credibility and veracity of information.

## Limitations

The necessary conceptual and temporal boundaries that constitute the framework of this review can also be viewed as its limitation. By focusing on a specific type of online misinformation—fake news—we intentionally excluded other variations of deceitful messages that can be influential in the public sphere, such as rumors, hoaxes, conspiracy theories, etc. This focus on the relatively recent species of misinformation led us to apply specific criteria to the stimulus material, as well as to limit the search by the period beginning in 2016. Since belief in both fake news and adjacent genres of misinformation could be driven by same mechanisms, focusing on just fake news could result in leaving out some potentially relevant evidence.

Another limitation is related to our methodological criteria. We selected studies to review based on the experimental design. Yet, the evidence of how people interact with misinformation may also be generated from questionnaires, behavioral data analysis, or qualitative inquiry. For example, recent non-experimental studies reveal certain demographic characteristics, political attitudes or media use habits associated with increased susceptibility to fake news [77, 78]. Finally, our focus on articles published in peer-reviewed scholarly journals means that potentially relevant evidence that appeared in formats more oriented toward practitioners and policy-makers could be overlooked. Future systematic reviews can present a more comprehensive view of the research area by expanding their focus beyond the exclusively "news-like" online misinformation formats, relaxing methodological criteria, and diversifying the range of data sources.

## Author Contributions

**Conceptualization:** Kirill Bryanov, Victoria Vziatysheva.

**Data curation:** Kirill Bryanov, Victoria Vziatysheva.

**Formal analysis:** Kirill Bryanov.

**Funding acquisition:** Kirill Bryanov, Victoria Vziatysheva.

**Investigation:** Kirill Bryanov.

**Methodology:** Kirill Bryanov, Victoria Vziatysheva.

**Project administration:** Kirill Bryanov.

**Supervision:** Kirill Bryanov.

**Validation:** Kirill Bryanov.

**Visualization:** Kirill Bryanov, Victoria Vziatysheva.

**Writing – original draft:** Kirill Bryanov.

**Writing – review & editing:** Kirill Bryanov, Victoria Vziatysheva.

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
