## [Decision Letter · Decision Letter 0]

24 Mar 2021

PONE-D-21-00513

Determinants of individuals’ belief in fake news: A scoping review

PLOS ONE

Dear Dr. Bryanov,

Thank you for submitting your manuscript to PLOS ONE. After careful consideration, we feel that it has merit but does not fully meet PLOS ONE’s publication criteria as it currently stands. Therefore, we invite you to submit a revised version of the manuscript that addresses the points raised during the review process.

The expert Reviewers gave generally favorable opinions on the manuscript. However, revision is needed especially to better justify inclusion/exclusion of sources, structure and clarification of some methodological and discussion choices. 

We look forward to receiving your revised manuscript.

Kind regards,

Stefano Triberti, Ph.D.

Academic Editor

PLOS ONE

Journal Requirements:

2. Please ensure that you refer to Figure 1 in your text as, if accepted, production will need this reference to link the reader to the figure.

3. We note you have included a table to which you do not refer in the text of your manuscript. Please ensure that you refer to Table 2 in your text; if accepted, production will need this reference to link the reader to the Table.

Reviewers' comments:

Reviewer's Responses to Questions

**Comments to the Author**

1. Is the manuscript technically sound, and do the data support the conclusions?

Reviewer #1: Yes

Reviewer #2: Partly

2. Has the statistical analysis been performed appropriately and rigorously? 

Reviewer #1: N/A

Reviewer #2: N/A

3. Have the authors made all data underlying the findings in their manuscript fully available?

Reviewer #1: Yes

Reviewer #2: Yes

4. Is the manuscript presented in an intelligible fashion and written in standard English?

Reviewer #1: Yes

Reviewer #2: Yes

5. Review Comments to the Author

Reviewer #1: This is a very useful paper. It is also clear and well written. A pleasure to review.

I’m not familiar with the methodologies used to do systematic reviews, thus I cannot properly evaluate the rigor of their methodology. However, from what I understood, their methodology is sound and robust.

I have two minor comments and two personal comments. But the manuscript can be published as is.

Minor comments:

P24: “In sum, findings in this section suggest that the general warnings and non-specific rhetoric of “fake news” should be employed with caution so as to avoid the possible backfire effect.” - What do the authors mean by backfire effect? It is used in the literature to mean a lot of different things.

P26: “The line of research concerned with accuracy-improving interventions reveals limited efficiency of general warnings and Facebook-style tags and suggests that simple interventions embedded in news interfaces to prime critical thinking and exposure to news literacy guidelines can induce more reliable improvements while avoiding normatively undesirable backfire effects.” - Add commas (e.g. after “tags”) otherwise there is no space to breathe.

Personal comments:

Page 11:

The “truth bias” is, I believe, psychologically implausible (Sperber, D., Clément, F., Heintz, C., Mascaro, O., Mercier, H., Origgi, G., & Wilson, D. (2010). Epistemic vigilance. Mind & Language, 25(4), 359-393). However, it is, for some reason, very influential in the literature on misinformation. The authors’ result regarding the absence of truth bias could be mentioned in the discussion. Indeed, this implausible assumption could influence the way researchers design their experiments and frame their results.

On the other hand, the “deception bias” makes sense in light of what we know about trust in the digital age: the problem is not that people trust fake news sources too much but rather that they don’t trust good sources enough (e.g. Fletcher, R., & Nielsen, R. K. (2019). Generalised scepticism: how people navigate news on social media. Information, Communication & Society, 22(12), 1751-1769 ; Altay, S., Hacquin, AS. & Mercier, H. (2020) Why do so Few People Share Fake News? It Hurts Their Reputation. New Media & Society; Pennycook, G., & Rand, D. G. (2019). Fighting misinformation on social media using crowdsourced judgments of news source quality. Proceedings of the National Academy of Sciences, 116(7), 2521-2526.; Mercier, H. (2020). Not born yesterday: The science of who we trust and what we believe.).

Reviewer #2: Review of PONE-D-21-00513

Reviewers: Stephan Lewandowsky and Muhsin Yesilada

Overall, the paper has clear importance; identifying the determinants of fake news beliefs can have useful implications for targeted interventions. The authors mapped out factors that could affect the outcome variables set out in the included studies. These factors were: message-level factors, individual-level factors, and intervention & ecological factors (although it was hard to determine how they identified these three factors).

The research team decided to use precedents to guide their scoping review (such as the PRISMA review guidelines). Still, there were major issues with the presentation of evidence. For example, certain aspects of the methodology would have benefited from being in the results or even discussion section. There was also a lack of comprehensive coverage of certain research areas (such as fake news interventions). These issues are described below.

Overall, the submission requires major revision before a favourable opinion can be given.

Major Points:

1. The number of studies included in the scoping review that investigate fake news interventions was limited. The intervention section highlights the prominence of inoculation-based research in this context; however, we noticed some studies that could have been included but that were not (e.g., Basol, Roozenbeek, & van der Linden, 2020; Roozenbeek & van der Linden, 2019). These studies should be included to provide a comprehensive overview of the research area.

2. The discussion needs further organization into subsections to make it clear to the reader where to locate information. There is no real conclusion subsection which makes it difficult to tie together the report's implications and findings. Also, much of the methods section (page 7-9) seems to assess and evaluate the included studies' methodological decisions rather than describing the review's methodology. Although this information is valuable, it is perhaps better suited for the results or discussion section.

3. It is not entirely clear how the research team identified the three-factor groups (message-level factors, individual-level factors, and intervention factors). Were these groups based on a precedent, or is there consensus in the research area that factors can be categorized into these three groups? It is important to have this information to justify the methodology and determine if any potentially important factors were missed. Also, the intervention factor group is paired with ecological factors - it is not entirely clear what the research team means by "ecological factors".

4. Table three, which sets out the key methodological aspects and results of the included studies, needs more procedural information. At present, it is not easy to interpret how the included studies might have arrived at their results. This information could provide a more comprehensive summary of the research area, particularly for people who might want to know more about the commonalities amongst procedures across studies.

Detailed Comments

[page]:[para]:[line]

2:1:3 The end of this sentence requires a citation. A study or report that has mapped out propaganda, misinformation, and deception in the public sphere over time would be relevant to cite here.

2:1:6-7 The authors state, "a lack of consensus over the scale and persuasiveness of the phenomenon". However, it is not clear what they are referring to. We assume they are discussing the persuasiveness of misinformation in general. It is also unclear how the references they cited support such a claim.

3:1:1-5 The authors refer to a "massive spread of politically charged deception". To a certain extent, the word massive is subjective and does not point to the problem's true extent. With that in mind, some statistics or figures would be helpful.

3:1:7 The authors note the "hypothesized societal effects" of deceitful messages; however, they do not explain what these societal effects might be. There are some studies out there that have investigated the causal effects of misinformation. These studies might be a good idea to cite to set the scene for the study. See below for a selection:

Schaub, M., & Morisi, D. (2020). Voter mobilisation in the echo chamber: Broadband internet and the rise of populism in europe. European Journal of Political Research, 59(4), 752–773. https://doi.org/10.1111/1475-6765.12373

Bursztyn, L., Egorov, G., Enikolopov, R., & Petrova, M. (2019). Social media and xenophobia: evidence from Russia (No. w26567). National Bureau of Economic Research.

Allcott, H., Braghieri, L., Eichmeyer, S., & Gentzkow, M. (2020). The welfare effects of social media. American Economic Review, 110(3), 629–76. DOI: 10.1257/aer.20190658

3:2:4 Citation needed to support this claim.

3:2:6 The authors refer to a "focal issue", but it is not entirely clear what the focal issue is.

4:1:5 The Sentence started with "because", but because of what? Consider re-writing for clarity.

4:3:2 The authors use the term "inductively developed framework", it would be good if the authors described what this means.

5 The eligibility criteria would benefit from being placed in a table. At present, the criteria are embedded in the text, and it does not make it easy for the reader to identify the information.

5:1:2 The time frame for the included studies should be explained. We assume the time frame for the included studies starts in 2016 to coincide with the initial Trump presidential campaign, but this an assumption. Further clarification would help.

5:1:7-12 This Sentence is far too long, consider rewording or breaking it down into several sentences.

5:1:12-14 The search started from studies already known to the researchers; cite this here for clarity.

5:2 This sentence is too long and would benefit from shortening. Also, the paragraph states that the trio of databases would most likely yield the most comprehensive results - but why? This needs to be clearly explained.

6:3:4 The authors do not explain why they chose these outcome variables.

8:1:1 The authors wrote, "As visible from the table", but do not state which table they refer to.

10:1 This paragraph would be a good place to explain how the factor groups were identified.

10:2:1 Avoid using rhetorical questions.

11:1:3-5 This Sentence is wordy and unclear; consider re-writing.

11:2:10-12 It is unclear what these numbers mean concerning the scale.

16:1:3 The authors note that "two major message level factors stand out"; however, it is not explained why these two stand out in particular.

17:1:1-5 This sentence is too long, consider rewording or breaking it down into several sentences.

17:1:7-10 This sentence is not clear on its own. Another sentence is needed to explain why these methodological differences lead to differing results.

17:1:16-18 It is argued that the differences in findings might be down to different study design choices - however, this needs more unpacking. The sentence alone does not explain why.

17:1:1-5 sentence is too long; consider rewording.

19:2:2 The authors use the terms "vary dramatically"; however, it is unclear what this means exactly; some figures or further quantification would be handy here.

19:2:8 The authors note an apparent limitation, but further explanation is needed to determine why it is a limitation.

19:3:3-4 Citation needed.

19:3:4-6 Citation needed.

20:2:5-7 The authors state that the correlations are statistically significant but do not provide an indicator of significance.

23:1:2 Citation needed.

23:2:2-5 Citation needed.

23:3:1 What were the guidelines?

23:3:11-12 "2.8 times less people were willing to share fake news following the treatment than before the treatment." - It is not clear what this statistic means and how it was identified.

24:3:9-10 What are the flags? We assume they are materials in a study but this is not entirely clear.

25:2:1-3 The authors discuss avoiding backfire effects, but research surrounding backfire effects is complicated. The current understanding is that backfire effects are not nearly as much of a concern as once thought - these recent findings should be reflected in this paragraph. (e.g., see Swire et al., 2020, DOI: 10.1016/j.jarmac.2020.06.006).

27:2:2 What is meant by a "common decision environment"? A definition here would be useful.

6. PLOS authors have the option to publish the peer review history of their article (what does this mean?). If published, this will include your full peer review and any attached files.

Reviewer #1: **Yes: **Sacha Altay

Reviewer #2: **Yes: **Muhsin Yesilada and Stephan Lewandowsky

---

## [Author Response · Author response to Decision Letter 0]

14 May 2021

Dear Drs. Altay, Lewandowsky, and Yesilada,

We are extremely grateful for your insightful and generously detailed feedback to our work. Based on your comments and suggestions, we have introduced some major changes to our report’s structure and evidence presentation. We believe that what resulted from this collaborative effort is a significantly improved manuscript. Our detailed responses to your comments, in the order we have received them, are listed in the table that can be found in an enclosed file entitled Response to Reviewers. We hope that you will find these responses sufficient.

Kind regards,

The authors.

---

## [Decision Letter · Decision Letter 1]

1 Jun 2021

PONE-D-21-00513R1

Determinants of individuals’ belief in fake news: A scoping review

PLOS ONE

Dear Dr. Bryanov,

Thank you for submitting your manuscript to PLOS ONE. After careful consideration, we feel that it has merit but does not fully meet PLOS ONE’s publication criteria as it currently stands. Therefore, we invite you to submit a revised version of the manuscript that addresses the points raised during the review process.

Some minor modifications have been suggested by the previous Reviewers. I believe these could be added in short time to improve the completeness of the manuscript. 

We look forward to receiving your revised manuscript.

Kind regards,

Stefano Triberti, Ph.D.

Academic Editor

PLOS ONE

Journal Requirements:

Reviewers' comments:

Reviewer's Responses to Questions

**Comments to the Author**

1. If the authors have adequately addressed your comments raised in a previous round of review and you feel that this manuscript is now acceptable for publication, you may indicate that here to bypass the “Comments to the Author” section, enter your conflict of interest statement in the “Confidential to Editor” section, and submit your "Accept" recommendation.

Reviewer #1: All comments have been addressed

Reviewer #2: All comments have been addressed

2. Is the manuscript technically sound, and do the data support the conclusions?

Reviewer #1: Yes

Reviewer #2: Yes

3. Has the statistical analysis been performed appropriately and rigorously? 

Reviewer #1: Yes

Reviewer #2: N/A

4. Have the authors made all data underlying the findings in their manuscript fully available?

Reviewer #1: No

Reviewer #2: Yes

5. Is the manuscript presented in an intelligible fashion and written in standard English?

Reviewer #1: Yes

Reviewer #2: Yes

6. Review Comments to the Author

Reviewer #1: The authors did a great job at addressing my (minor) comments, I am now satisfied with the manuscript.

It's too bad that the very small scale of the fake news problem is not mentioned in the introduction (e.g. Allen et al. 2020) but the article is, I believe, good enough to be published as is.

Finally I would like to thank the authors for their work, it's a very useful paper!

Reviewer #2: Review of MS PONE-D-21-00513-R1

by Bryanov & Vziatysheva

Reviewer: Stephan Lewandowsky

Summary and Overall Recommendation

The paper is clearly important; identifying the determinants of fake news beliefs can have implications for successful interventions. The authors mapped out factors that could affect the outcome variables set out in the included studies. These factors were: message-level factors, individual-level factors, and intervention & ecological factors (although it was hard to determine how

they identified these three factors).

I reviewed the paper at the previous round (together with a PhD student whom I did not consult at this round to save time). Our judgment was positive in principle, but we requested major revisions, in particular relating to (1) the small number of studies; (2) clarity of the discussion and (3) the three factors being identified; and (4) expansion of the main Table (Table 3 in the original submission).

The revision has addressed these points and I found the manuscript to be much improved and (nearly) ready for publication, subject ot the minor comments below.

Detailed comments

[line#]

165 I am not entirely clear why “cognitive science research on false memory recognition” would be “obviously irrelevant”?

173 Does “non-experimental” mean the authors excluded correlational studies? I would have thought most individual-differences research may involve correlational studies that do not include an experimental intervention. Perhaps the authors mean “non-empirical”? If they did exclude correlational studies I would be curious to know why.

23 The authors may be interested in DOI 10.3758/s13421-019-00948-y as another demonstration of social influences (although it is only indirectly related to fake news because the study compares pro- and anti-science blog posts).

387 Insert paragraph break before “As this…”.

392-394 This sentence is ungrammatical.

504 “the Indian sample” pops out of nowhere—this deserves a bit more explanation. Why India? What can be learned from this?

579 “messages” should be plural?

7. PLOS authors have the option to publish the peer review history of their article (what does this mean?). If published, this will include your full peer review and any attached files.

Reviewer #1: **Yes: **Sacha Altay

Reviewer #2: **Yes: **Stephan Lewandowsky

---

## [Author Response · Author response to Decision Letter 1]

5 Jun 2021

Dear Drs. Altay and Lewandowsky,

We are grateful for your continued contributions to the improvement of our work. The revised manuscript addresses each comment you have raised in the latest round of review. Further details on the changes we have made can be found in the table appended to the enclosed file, titled Response to reviewers. We hope that you will deem the resulting manuscript fit for publication.

Kind regards,

The authors.

---

## [Editor Report · Decision Letter 2]

11 Jun 2021

Determinants of individuals’ belief in fake news: A scoping review

PONE-D-21-00513R2

Dear Dr. Bryanov,

We’re pleased to inform you that your manuscript has been judged scientifically suitable for publication and will be formally accepted for publication once it meets all outstanding technical requirements.

Kind regards,

Stefano Triberti, Ph.D.

Academic Editor

PLOS ONE
---

## [Editor Report · Acceptance letter]

16 Jun 2021

PONE-D-21-00513R2 

Determinants of individuals’ belief in fake news: A scoping review Determinants of belief in fake news 

Dear Dr. Bryanov:

I'm pleased to inform you that your manuscript has been deemed suitable for publication in PLOS ONE. Congratulations! Your manuscript is now with our production department. 

Kind regards, 

on behalf of

Dr. Stefano Triberti 

Academic Editor

PLOS ONE